# Assessment of Solar Dehumidification Systems in a Hot Climate

Kheira Anissa Tabet Aoul , Ahmad Hasan * and Joud Al Dakheel

Architectural Engineering Department, College of Engineering, United Arab Emirates University,
P.O. Box 15551 Al Ain, UAE; kheira.anissa@uaeu.ac.ae (K.A.T.A.); 200935298@uaeu.ac.ae (J.A.D.)
* Correspondence: ahmed.hassan@uaeu.ac.ae; Tel.: +97-155-545-4069

**Abstract:** Solar thermal-powered desiccant dehumidification systems are attracting attention for cooling load-dominated climates. However, their performance varies substantially from place to place depending on climatic conditions, which therefore warrants a tailored design and specification at each geographical location. The current article attempted to investigate the feasibility of extending an existing solar thermal system in a school building in Abu Dhabi to provide dehumidification for the existing air condition system through a desiccant system. The system performance was predicted through a Transient System (TRNSYS) Simulation model to determine the energy savings achieved by the solar-assisted dehumidification system. The current articles determined the effect of fluid flow rate, solar radiation concentration, and heat exchanger effectiveness at the dehumidification of the fresh air as well as energy saved by the proposed system. It was concluded that the system can remove 35% moisture from the air, simultaneously saving 10% of the building's energy. The system cost and benefit analysis revealed a payback period of 7.5 years, considered slightly higher for an attractive investment in such systems.

**Keywords:** desiccant dehumidification; solar thermal; TRNSYS; hot climate; architectural integration; economics analysis

## 1. Introduction

The building sector accounts for around one-third of global greenhouse gas emissions and more than 40% of global energy demand [1]. In the United Arab Emirates (UAE), buildings consume about 90% of the total electricity used in the country [2]. Abu Dhabi, the capital of the UAE, experienced a sharp increase in its energy consumption from 65,000 GWh to 110,000 GWh between 2010 to 2018, with a monotonic increase in installed electrical power capacity from 71,000 GWh to 119,000 GWh (Figure 1) [3]. About 32% of the building sector's electricity consumption is attributed to air-conditioning systems worldwide [4,5]. Comparatively, over 70% of the electricity is used to meet the air-cooling demand in Abu Dhabi, given its hot climatic conditions [6]. Recently, the electric power consumption has been closely approaching the power production capacity, leaving an excess production of only 9100 GWh in 2018, as shown in Figure 1 [3]. The production–consumption data, on one hand, show little idle running, but on the other hand, increase the risk of reduced surplus at peak time. The increasing energy consumption may further reduce the demand–supply gap and the power network may be at risk of peak-time power shortage. In added power production capacity, several parallel attempts are required on demand-side management to reduce energy consumption to maintain power grid stability.

The fact that the domestic and commercial building sector accounts for 78% of the total electrical energy consumption in the UAE [7], as shown in Figure 2, naturally draws a focus to adopting measures to reduce building energy consumption and looking for an alternative, less energy-consuming building services system. In this line, the UAE is aiming to amplify power generation from clean energy to 30% by 2030, where a sizable 25% to 30% of its electricity needs will be from both nuclear and solar energy [8]. The latter is facilitated by the abundance of solar radiation, with an annual average global solar irradiance is

6 kW h/m$^2$/day [9] Throughout the year, the outdoor relative humidity remains above 60% in Abu Dhabi, which eventually leads a higher yearly energy consumption of 57% in dehumidification, compared to 43% in cooling in a specific building [10]. In a bid for a less energy-intensive system adaptation in buildings, an alternative way of dehumidification system remains relatively unexplored in UAE.

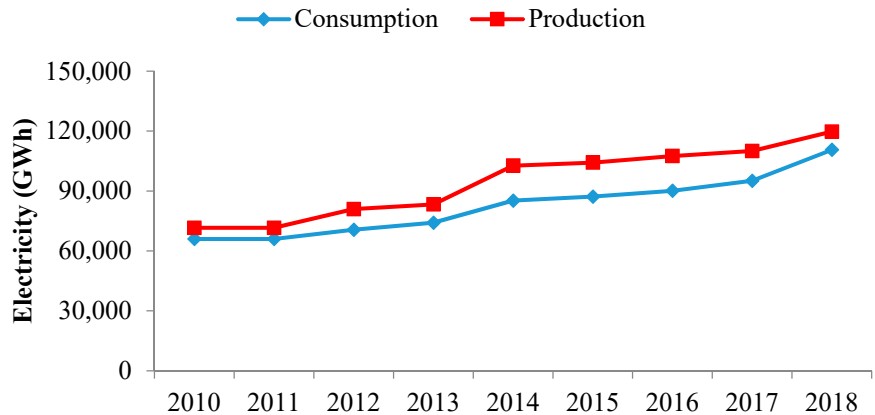

**Figure 1.** Comparative propagation of electricity production and consumption in Abu Dhabi, United Arab Emirates (UAE) [3].

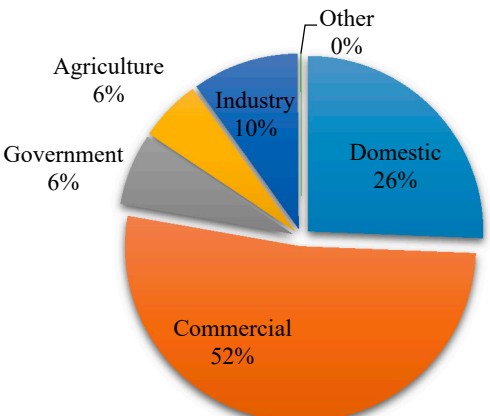

**Figure 2.** Electricity consumption in different sectors of Abu Dhabi, United Arab Emirates (UAE) [11].

A solar-assisted desiccant dehumidification system offers an advantage over electrical heating-based desiccant systems in the UAE, having a higher solar radiation flux of 2370 kWh/m$^2$-year on an inclined plane [5,12]. The conventional method of dehumidification relies on employing condensing coils to drain humidity from cooling air streams consuming on average 30% cooling energy consumption in UAE [6,13]. The energy-intensive dehumidification system needs to be replaced by an energy-efficient dehumidification system specifically suited for the hot climate of the UAE [14]. The solar-assisted dehumidification systems have been proven effective for hot and humid climates for large-scale building air conditioning applications [15]. Zhao et al. [16] led an experimental investigation on a desiccant dehumidification unit using silica gel coating and found that the unit can dehumidify 100% fresh air under mild humidity loads. Hao et al. [17] investigated the performance of a solar-assisted desiccant dehumidification system coupled with a chilled roof and displacement ventilation and reported up to 8% energy savings compared to a condensing coils-based dehumidification system. Solar-assisted desiccant dehumidification systems were further improved by employing nanofluids for enhanced heat transfer applied in a greenhouse buildings in Saudi Arabia [18]. In another study done in Saudi Arabia, the performance of an integrated evaporative-cooled window with

a desiccant dehumidification system combined with a photovoltaic/thermal (PV/t) system was examined [19]. The hybrid system showed a reduction in the inner window temperature by 5 °C to 7 °C, resulting in an 11% decrease of the total cooling load during the summer months. In another study, solid desiccant dehumidification employing photovoltaic thermal (PV/T) for thermal energy supply with Maisotsenko coolers was experimentally evaluated in the UAE, employing a PV/T with a coverage area of 681.0 m², which provided the 36.5% thermal energy required for desiccant regeneration, thus saving 139.7 MWh per year electrical energy [20]. The solar-assisted dehumidification systems have been studied in Italy and were found effective to provide environmentally friendly comfort conditions [21]. These brief literature findings provide insight into the technical feasibility of the desiccant dehumidification system in hot climate zones. However, the system has not been studied comprehensively in terms of influencing operating parameters. The previous research article highlights the need for more comprehensive studies into the solar-assisted dehumidification system specific to the UAE climate. The current article starts investigating certain aspects of solar-assisted dehumidification in the UAE and will generate further insight into the topic through more research in the future.

Thus, the current article aimed to investigate the possibilities of optimizing operating parameters to determine the most feasible operational modes of the solar-assisted desiccant dehumidification system in the context of the UAE in specific and in the hot and humid climatic zone in general. The TRNSYS simulation program was employed to model the system leading to an assessment of energy and economic savings of the proposed system.

### 1.1. Building Details

A school building located in Abu Dhabi and compliant with Estidama, the green building code applicable in the UAE was selected to be examined for this research. The school has an existing rooftop solar thermal collector array covering 46 m² roof space. The array comprises 18 solar thermal collectors with a 2.5 m² aperture area of each collector connected to a central hot water storage tank to provide hot water for the swimming pool, kitchen, and showers. The collector system produces 117,760 kWh thermal energy, thus saving 4.5% of building annual energy demand. Although the solar thermal collector is very efficient during wintertime serving hot water needs, the system remains idle during summer months (April–October) as the hot water demand during this time drops to nil. To extend the use of the system all-year-round, the authors came up with the idea of connecting the solar thermal system to a desiccant dehumidification system to provide the heat required for desiccant regeneration in the summer months, to lower the energy consumption due to air conditioning. The scope of the work was defined by simulation studies involving system capacity assessment and optimization in the climatic conditions of the UAE.

The sprawling school building comprises two floors with a 14,694 m² footprint area. Apart from classrooms, the school consists of offices, auditoriums, outdoor spaces, playgrounds, gymnasiums, cafeterias, toilets, and other support facilities. The building is oriented towards the northwest direction in the longest (or shortest) side of its rectangular form as shown in Figure 3.

The building employs a conventional chilled water air conditioning system and its current air dehumidification is achieved through conventional condensing coils. The latter consumes 50% extra energy compared to solar desiccant dehumidification systems [15], thus offering an alternative option for dehumidification. The overall building energy performance of the school, shown in Table 1, is based on the provided simulation data shown in Table 1.

Simulated energy consumption data employing typical meteorological year (TMY) was collected from the school to serve as a baseline case presented in Table 2.

The installed solar thermal system contributed 117,760.00 kWh out of 231,840 kWh of water heating and 2,088,493.2 kWh of the total annual energy consumption of the building.

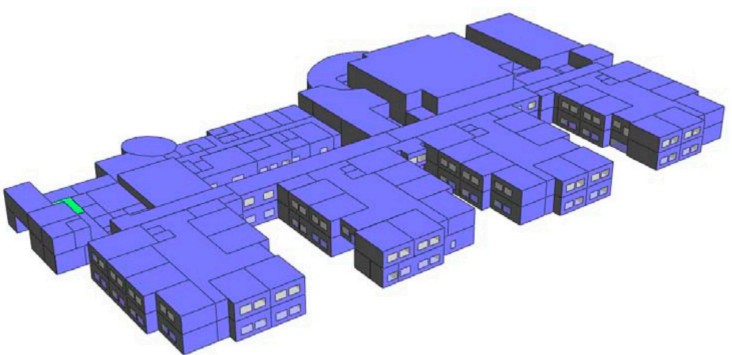

**Figure 3.** Building form of the school.

**Table 1.** Baseline data of the school building.

| Sectors | Baseline Value |
|---|---|
| Building Annual Energy Consumption (kWh) | 2,206,253 |
| Peak Demand (kW) | 1392.9 |
| Annual Renewable Energy Generation (kWh) | 117,760.0 |
| Percentage of Energy from Renewables | 5.3% |
| Occupied Hours Per Day (hrs) | 9 |

**Table 2.** The building energy consumption for a typical year.

| Sector | Peak Demand (kW) | Annual Consumption (kWh) |
|---|---|---|
| Space Cooling | 360.5 | 777,607.7 |
| Heat Rejection | 49.4 | 151,257.6 |
| Space Heating | 79.7 | 66.0 |
| Pumps | 42.8 | 105,742.8 |
| Fans—Interior | 304.6 | 276,997.0 |
| Interior Lighting | 188.3 | 410,262.4 |
| Service Water Heating | 200.0 | 231,840.0 |
| Receptacle/Process Equipment | 167.5 | 252,479.7 |
| Total | 1392.9 | 2,206,253.2 |

### 1.2. Solar-Assisted Dehumidification Challenges

Temperature, humidity, and extreme solar radiation intensity are the most influential climatic challenges in the UAE. The National Bureau of Statistics reported the maximum annual temperature average high of 50.2 °C in July, while with an average low of 15 °C in January, the peak average relative humidity of 84%, and the annual solar radiation intensity of 2370 kWh/m$^2$ [12]. Given the hot and humid climate of Abu Dhabi, extensive dehumidification is required during both summer and winter seasons, although the cooling load in winter drops but remains negligible. The dehumidification needs to warrant a year-round operation of a cooling system, which otherwise can be shut down in winter, provided that an alternative dehumidification system is incorporated in the building ventilation scheme.

### 2. Methodology

The solar-assisted desiccant system integrated with the school building was simulated using the TRNSYS simulation program to predict the dehumidification and energy performance of the system by applying Abu Dhabi TMY weather data.

The developed system in TRNSYS consists of three fluid flow loops, as shown in Figure 4 as a scematic diagram and the detailed components in the Appendix A. The first loop circulates water between the concentrated thermal collector (Type1245-2), water storage tank (Type60u), and the heat exchangers (Type91b). The collector heats water to a required temperature (set point 90 °C) that passes through the storage tank to the heat exchanger to transfer heat to returning air. The second flow stream flows fresh air from outdoors through the desiccant system to remove humidity, through a heat recovery wheel

to pre-cool the air, and delivers it to the building. The third flow stream consists of return air from the building passing through the heat recovery wheel to recover cooling energy contained in return air, a heat exchanger to heat the returned less humid air, which is then passed through the desiccant wheel to remove the moisture contained in the wheel, so-called desiccant wheel regeneration.

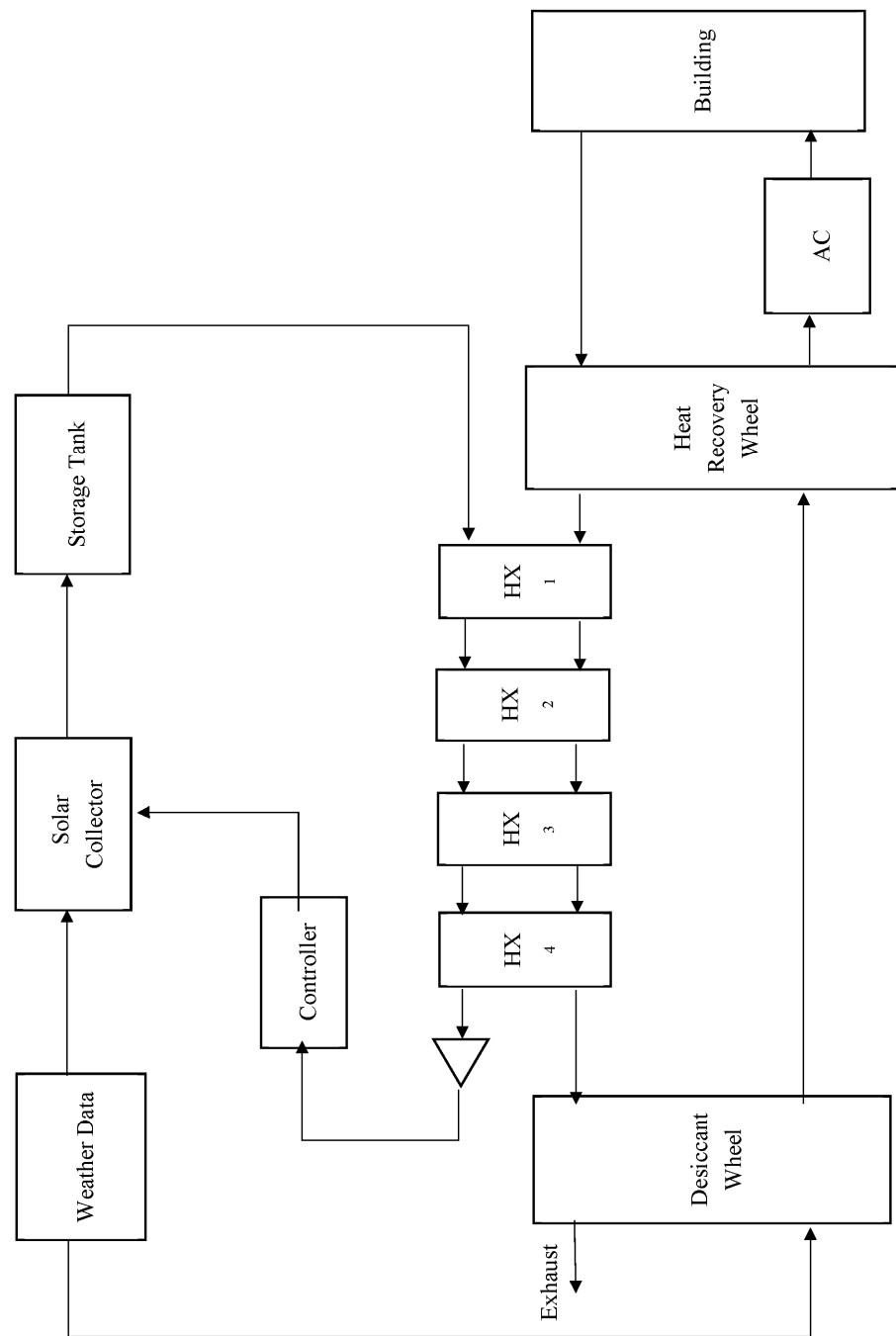

**Figure 4.** TRNSYS model of the solar-assisted desiccant cooling system.

The model computes system performance by considering the Perez sky model for diffuse radiation (Type 15-6) at inclined thermal collectors at the latitude angle. A two-axis radiation tracking is imposed on the inclined surface at latitude under solar radiation concentration to gain maximum solar radiation intensity. The radiation is computed on the sloped surface (collector) considering 0.2 ground reflectance. The computed inclined

surface incident radiation (*Iin*) enters the aperture through area (*Aa*) of the compound parabolic concentrating collector. At this point, the radiation per unit area being incident on the absorber is increased by a factor of geometrical concentration defined by concentration ratio (*CR*) as follows

$$CR = \frac{A_a}{A_s} = 2\left(\frac{\bar{x}}{s}\right)cos\theta_c - \left(\frac{\bar{x}}{s}\right)^2 \frac{Sin\theta_c}{Sin\theta_c + 1} + sin\theta_c - cos^2\theta_c. \tag{1}$$

The reflective radiation loss through walls of the trough are accounted for by effective reflectance

$$\rho_{eff} = \frac{I_R}{I_{in} \cdot CR}. \tag{2}$$

The effective reflection has been analyzed Rabl as

$$\rho_{eff} \approx \rho R^n \tag{3}$$

where R represents the wall reflectance and n denotes the number of internal reflections. It has been shown that both n and CR do not depend on the width of the collector (s). The thermal performance is modeled using the Hottel–Whillier equation

$$Q_u = \frac{A\,F_R}{N_S}\sum_{j=1}^{N_s}\left(l_{in}(\tau\alpha) - U_{L,j}(T_{i,j} - T_a)\right) \tag{4}$$

The overall transmittance–absorptance products are calculated as:

$$(\tau\alpha) = \frac{I_bT(\tau\alpha)b + I_bT(\tau\alpha)d}{I_T} \tag{5}$$

The collector absorptance ($\alpha$) is taken as 0.8 and transmittance ($\tau$) as 0.7 with a bottom loss coefficient (*UL*) of 0.83 W/m²K. A series connection is imposed for the 18 thermal collectors (Type 1245) with an aperture area of 2.5 m² each, employing water as heat transfer fluid with a water flow rate ranging from 1000 kg/h to 5000 kg/h The collector outlet temperature (*Ti,j*) and the useful thermal energy gain (*Qu*) is monitored for the specified simulation time duration as an indicator of the collector performance. The hot water produced at temperature (*T*) is fed into a stratified hot water storage tank (Type60u) with an internal heat exchanger and two embedded auxiliary heaters at upper middle and lower middle locations. Both the auxiliary heaters setpoint temperatures are made in line with the required water temperature of 90 °C to trigger auxiliary heating in case the collector outlet temperature cannot maintain the desired temperature. The auxiliary heaters are operated by a master–slave control scheme. The natural convection heat loss to ambient air ($h_o$) is calculated employing:

$$h_\circ = \frac{Nu_D k}{d_\circ} \tag{6}$$

where: $Nu_D$ is the local Nusselt number given by:

$$Nu_D = CRa^n$$

where *Ra* is the Ryleigh no. and *C* is the film coefficient. For a typical flat plat collector *C* is around 0.54 and that of n is approximately 0.25 [22].

Both the hot side as well as cold side flow rates are kept the same to mimic a continuous flow system. After assuring the desired temperature, hot water is passed on to four parallel

flow zero capacitance, constant effectiveness heat exchangers (Type 91-b-2) connected in series. The effectiveness of the heat exchanger is calculated from the equation:

$$\varepsilon = \frac{1 - \exp\left(-\frac{UA}{C_{min}}\left(1 + \frac{C_{min}}{C_{max}}\right)\right)}{1 + \frac{C_{min}}{C_{max}}} \tag{7}$$

Under this scheme, the maximum heat transfer rate is calculated through the hot side (hot water) and cold side (return air) temperature difference, while the heat exchanger effectiveness ($\varepsilon$) and overall heat transfer coefficient ($U$) are provided as a parameter with $\varepsilon$ varying from 0.4–0.8 and U being constant at 2.77 W/K. Eventually, the fluid outlet temperatures are measured leaving hot water and return air. Finally, the return air is passed across the silica gel-based rotary desiccant dehumidifier wheel working on the input humidity ratio mode. The desiccant dehumidifier has a rated power capacity of 6 kW with a supply airflow rate of 1000 kg/h and returns an airflow rate of 500 kg/h. The desiccant wheel dehumidifies the inlet fresh air with the thermal energy provided by the return air stream.

*Model Inputs*

The TRNSYS model is prepared considering crucial parameters of each component of the desiccant dehumidification system as shown in the Table 3.

**Table 3.** Summary of crucial input parameters for each component of the desiccant dehumidification system considering all inputs in SI units.

| Parameter | Fixed Value | Variable Range |
|---|---|---|
| **Thermal Collector** | | |
| Collector fin Efficiency factor | 0.7 | |
| Bottom, edge loss coefficient | 3.0 | |
| Absorber plate emittance | 0.7 | |
| Absorptance of absorber plate | 0.8 | |
| Cover refractive index | 1.526 | |
| Extinction coeff. thickness product | 0.026 | |
| Inlet temperature | | Default from weather data |
| Concentration Ratio | | 1–5 |
| Collector slope | 24 | |
| Fluid Specific heat | 4.190 | |
| Fluid density | 1000 | |
| Tank loss coefficient | 3.0 | |
| Cold side flowrate | 5000 | |
| **Heat Exchanger** | | |
| Specific heat of HTF | 4.19 | |
| Source side flowrate | | 1000–5000 |
| Load side flow rate | 500 | |
| Overall heat transfer coefficient of exchanger | 10 | |
| Heat Exchanger effectiveness | | 0.5–0.8 |
| **Desiccant wheel** | | |
| Regeneration air flowrate | 500 | |
| Rated Power | 671.1 | |
| **Pump** | | |
| Water flowrate | | 1000–5000 |
| Rated power | 2684 | |
| Motor efficiency | 0.90 | |
| **Fan** | | |
| Air flowrate | 500 | |
| Fluid specific heat | 4.190 | |
| Maximum power | 60 | |
| Conversion coefficient | 0.05 | |
| Power coefficient | 0.5 | |
| **Building Ventilation** | | |
| Inlet mass flow rate | 500 | |
| Thermal capacitance of zone | 24000 | |
| Moisture capacitance of zone | 200 | |
| Air temperature | | Default From weather data |
| Solar radiation intensity | | Default from weather data |
| humidity ratio | | Default From weather data |

## 3. Results and Discussion

### 3.1. Effect of Flowrate

The performance of a thermal collector strongly depends on the flow rate of heat transfer fluid provided *y* the Equation (8) below:

$$Q_{u=}\ \dot{m}C_p\left(T_{in}-T_{out}\right) \tag{8}$$

The effect of heat transfer fluid flow rate ($\dot{m}$) was studied on the fluid temperature at the outlet of the solar thermal collector ($T_{out}$) varying the flow rate from 1000 kg/h to 5000 kg/h gradually. The fluid temperature at the collector was plotted for the extreme summer and winter months as a function of the flow rate. It was observed that the fluid temperature in winter temperature lagged substantially over the summer temperature at a lower fluid flow rate, however the difference decreases with increasing flow rate. Eventually, a collector outlet temperature of 57 °C in winter and 80 °C in summer was achieved at 5000 kg/h fluid flow rate as shown in Figure 5, since the temperature achieved at the outlet of the collector was a strong function of fluid flow rate.

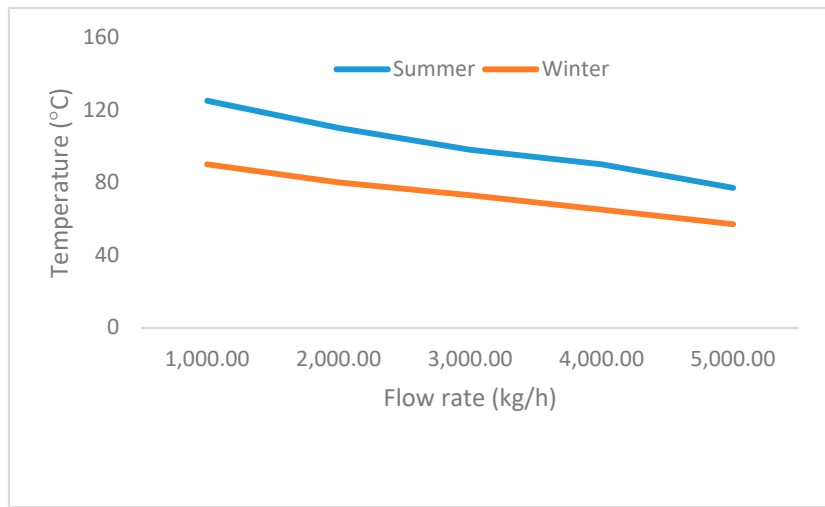

**Figure 5.** The collector outlet temperature as a function of fluid flow rate for summer and winter days in Al Ain, UAE.

### 3.2. Solar Concentration

The available radiation at the solar collector surface is a function of concentration ratio, as shown by the Equation (1). A higher CR achieves lower reflection losses ($\rho_{eff}$), which eventually achieves higher incident radiation ($I_{in}$), as shown by Equation (2). The higher $I_{in}$ eventually yields higher useful energy gain ($Q_u$), which is directly proportional to the collector outlet temperature ($T_{out}$), as given by the Equation (4). The conventional building applied solar collectors intended for space heating are required to heat the air around 25 °C, achievable with any collector in most of the climates. However, for a desiccant dehumidification system, air must be heated up to 75 °C to dry out the desiccant material up to 80%. Such a high air temperature can only be attained by concentrated collectors currently being employed in the research. However, the temperature attained depends largely on concentration ratio, the flow rate of heat transfer fluid, and the climatic conditions. Simulations were conducted to determine the optimal concentration ratio to achieve the desired temperature in the winter months (Dec. and Jan.) as well as in the summer months (June-July). An almost linear increase in collector outlet temperature was observed with increasing solar radiation concentration, as shown in Figure 6, in both summer and winter. It was observed that without solar concentration (CR = 1×), the water temperature at the collector outlet remained below 60 °C in winter, which cannot meet the

temperature demand of the desiccant regeneration. In winter, the temperature required for desiccant regeneration (80 °C) was achieved with the CR above 3×. On the contrary, the temperature required for dehumidification at the selected flow rate of 5000 kg/h was nearly achieved without concentration, however the temperature increased substantially reaching 94 °C at a concentration 3×. The temperature exceeded 100 °C at higher concentrations with the same flow rate.

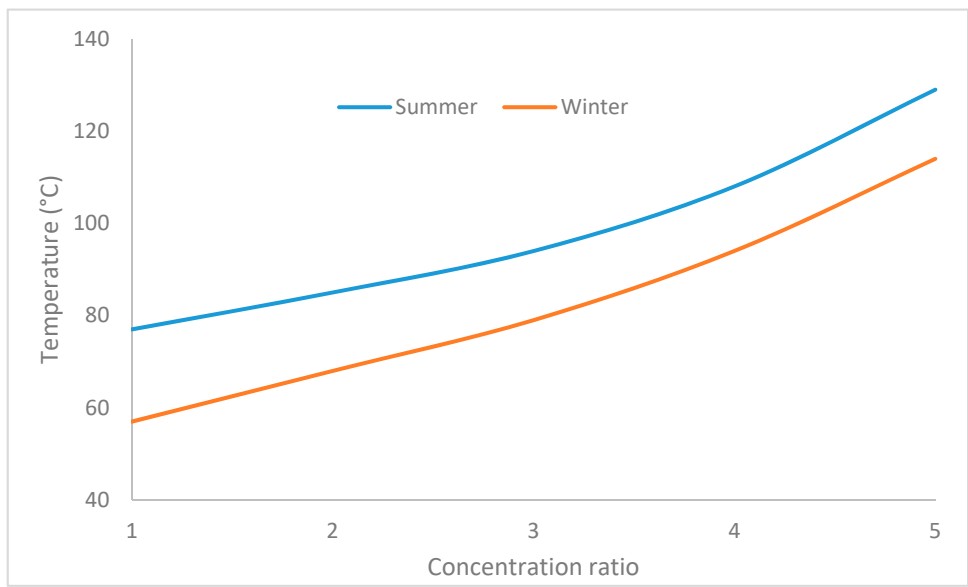

**Figure 6.** Temperature variation versus solar radiation concentration at a fixed fluid rate of 5000 kg/h.

The solar-assisted dehumidification system is therefore considered to be more effective at 3× meeting the minimum temperature requirement of 80 °C in winter and not exceeding the safety requirement of the collector (below 100 °C) at the prescribed flow rate. However, the system can be operated at a higher concentration once the flow rate is increased substantially beyond the selected flow rate. The rest of the simulations were conducted at a concentration ratio of 3×, with a flow rate of 5000 kg/h to study the influence of heat exchanger effectiveness and diverter mixing ratios on temperatures, energy transfer rates, relative humidity, and humidity ratios on yearly basis.

### 3.3. Fluid Outlet Temperatures

The system was simulated with CR 3 at 5000 kg/h flow rate to determine the temperature at key points for moderate months to determine the performance of the system on average based on Equations (1)–(4). Temperature values were predicted for both water streams (hot water as a heat source) and air stream (return air as a load side) to determine the effectiveness of the collector and heat exchanger during the heat transfer mechanism. The temperature monitoring locations were water temperature at the outlet of the solar thermal collector (Water-coll-out), water temperature at the inlet to heat exchanger (Water-in-HX), water temperature at the outlet to heat exchanger (Water-out-HX), return air temperature at the inlet to heat exchanger (Air-in-HX), and the return air temperature at heat exchanger outlet (Air-out-HX). The heated water at the collector outlet was passed on to the heat exchangers arranged in series to transfer heat to air, which eventually would dry out the desiccant wheel effectively at 75 °C. The water and air cycle in the heat exchanger are shown in Figure 7a,b. The water exit temperature served as a condition of minimum temperature to heat the return air since a temperature below this would not regenerate the desiccant wheel optimally [23]. The hot water flow rate, as well as the return air stream flow rate, was adjusted to provide maximum heat transfer from hot water (source side) to the return air (load side) to achieve maximum load side fluid outlet temperature. As

a condition, the source side water flowrate was dropped as much as possible to achieve a water exit temperature of 75 °C in order to achieve the maximum air temperature that could eventually dry out the desiccant wheel effectively, as shown in Figure 7a,b. It can be observed in Figure 7a that the collector outlet temperature reached 88 °C, which was slightly heated by the auxiliary heater in the storage tank to reach a set point inlet temperature of 90 °C to the heat exchanger (Water-in-HX). On other hand, the return air from the space entered the heat exchanger (Air-in-HX) at 36.7 °C. The heat exchangers arranged in series provided substantial stay time to transfer heat from the source side to the load side. Eventually, water exited the heat exchanger at 81.3 °C (Wate-out-HX), heating the return air to 63.7 °C (Air-out-HX). It can be observed that the return air being heated to 63.7 °C still was substantially below the required desiccant wheel regeneration temperature of 75 °C for effectively drying the desiccant wheel. As shown in Figure 7b, the return air heated by hot water was fed into the desiccant wheel (Return-air-in-DW) at 63.7 °C, which left the desiccant wheel at 60.9 °C (Return-Air-Out-DW). On the other hand, fresh air entered the desiccant wheel at as shown 39.8 °C (Fresh-Air-in-DW) and was heated to 45.6 °C (Fresh-Air-Out-DW). As the fresh air heats at the outlet of the desiccant wheel, it may limit the performance of the desiccant dehumidification system as it may increase the sensible cooling load of the building being dehumidified in summer. However, it can turn into a benefit in case the building needs to be heated in the winter months.

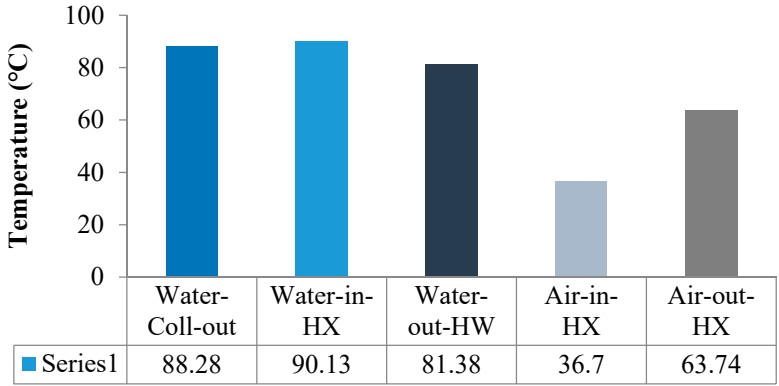

Water and Air flow into Heat Exchanger (HX)

(**a**)

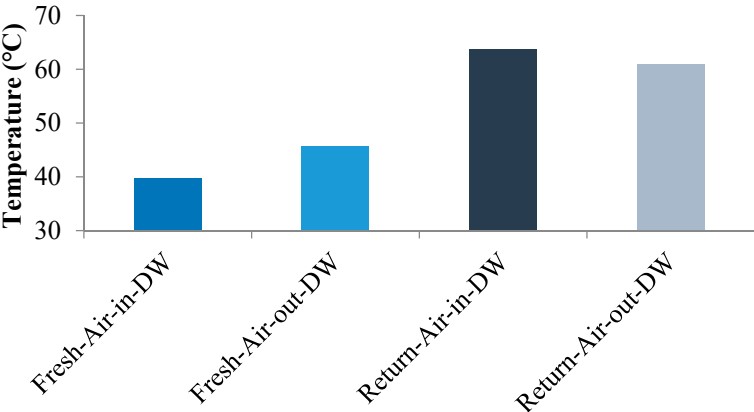

Air flow into Desiccant Wheel (DW)

(**b**)

**Figure 7.** Heat transfer fluid temperature for the optimum concentration ratio for (**a**) Heat exchanger and (**b**) desiccant wheel.

### 3.4. Collector Energy Contribution

The energy contribution of the solar thermal collector (Coll-Energy) to the energy provided to the return air for regenerating the desiccant wheel (Exch-Energy) defines the potential effectiveness of the collector. It can be noted that the collector provided 18,000 kWh to the energy exchanged to return air in the heat exchanger of 27,500 kWh, thus making a 65% contribution to total energy carried by return air for drying out the desiccant wheel (Figure 8). The rest of the 9500 kWh energy was provided by the auxiliary heater embedded within the hot water storage tank, so it can be concluded that the collector still provides a substantial amount of solar thermal energy required by the dehumidification system.

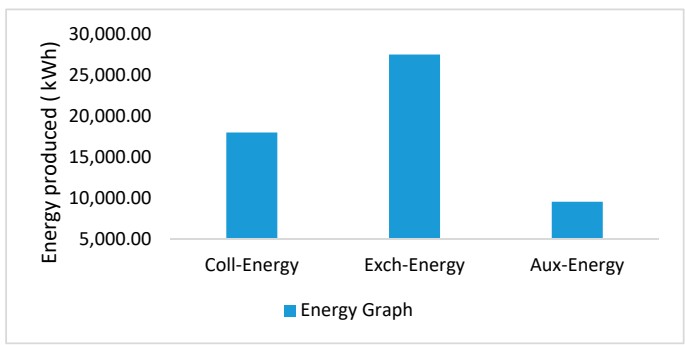

**Figure 8.** Energy contribution of the solar thermal collector to the total energy transferred to the return air stream.

### 3.5. Dehumidification Effect

The effectiveness of the solar-assisted dehumidification system is determined by comparing the humidity ratio (HR) at the inlet (HR-DX-in) to the same at the outlet (HR-DX-out) of the DX- wheel, which indicates the amount of moisture removed from the fresh air. It can be observed from Figure 9 that HR at the inlet to the DX wheel varied substantially between day and night from all the from day 1 till day 3, while the HR at the outlet to the DX wheel showed negligible variation. Figure 9 shows that the system was effective in maintaining the air leaving the DX wheel (HR-DX-Out) at a certain level of relative humidity. At peak time during the day, the HR dropped from 0.024 kg/kg to 0.016 kg/kg showing a drop of 35%.

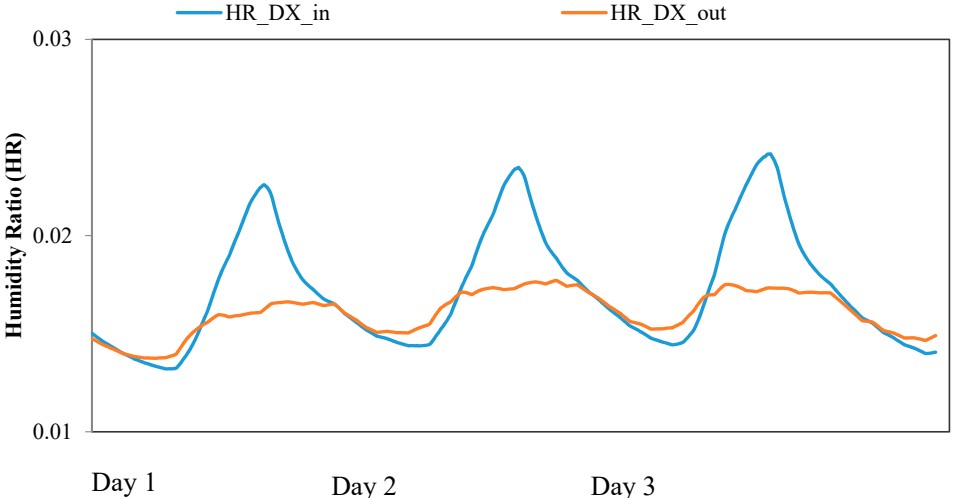

**Figure 9.** Humidity ratio of the fresh air at the inlet and outlet of the desiccant dehumidification wheel.

### 3.6. Heat Exchange Effectiveness

The effect of heat exchanger (HX) effectiveness at the heat exchange from source side (hot water) to the load side (return air), as provided in Equation (7), was studied by varying the effectiveness from 0.5 to 0.8 as shown in Figure 10a. The water temperature at the outlet (water-out-heat-exch) dropped from 86 to 77 °C, while at the same time the air temperature at the heat exchanger outlet let (Air-out-heat-exch) increased from 61°C to 71 °C when the HX effectiveness increased from 0.5 to 0.8, as shown in Figure 10a. It highlights that the heat exchanger effectiveness can substantially increase the heat transfer from the source side to the load side, helping to achieve the required temperature for desiccant wheel regeneration.

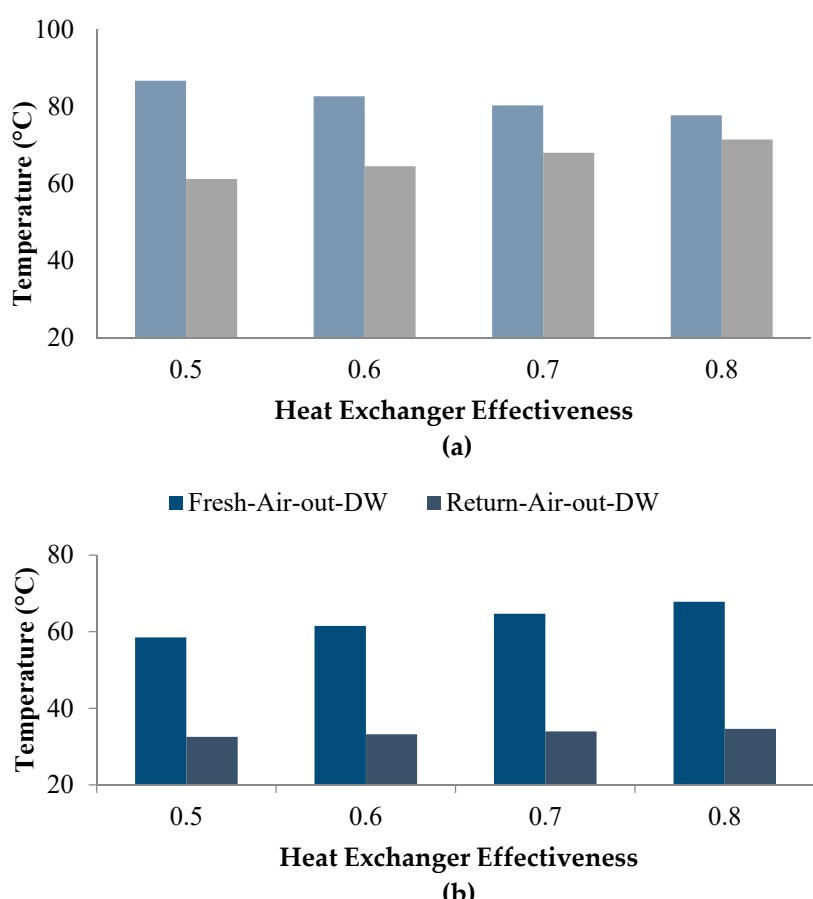

**Figure 10.** Temperature flow cycles at several levels of effectiveness for heat exchanger (**a**) and desiccant wheel (**b**).

At the same time, the fresh air temperature at the outlet temperature of the desiccant wheel (Fresh-Air-Out-DW) increased from 58 to 67 °C as shown in Figure 10b. The return air temperature increased slightly at the outlet of the desiccant wheel (Return-Air-Out-DX), increasing from 32 to 34 °C. It can be concluded that the heat exchanger effectiveness improves the heat transfer between air and water stream in the heat exchanger, as well between the return air and the desiccant wheel, which eventually can improve the desiccant heat regeneration.

### 3.7. Diverter Mixing Ratio

The third influencing parameter is the diverter mixing ratio, which is the ratio of return air entering each one of the four heat exchangers in order from the first heat exchanger to the last. Three arrangements were simulated labeled as X1 ( 0.4, 0.3, 0.2, and 0.1, which

means the first heat exchanger will take the maximum while the last heat exchanger will take the minimum amount of return air), X2 (0.25, 0.25, 0.25, and 0.25, meaning all heat exchangers will take the equal amount of return air), and X3 (0.1, 0.2, 0.3, and 0.4, which means the first heat exchanger will take the minimum while the last heat exchanger will take the maximum amount of return air) respectively for diverters 1, 2, 3, and 4. The heat transfer rate and the associated temperature rise of the return air were plotted as shown in Figure 11. The heat transfer rate of the exchanger increased from 7409 kWh to 31,571 kWh as the diverter ratio changed from X1 to X3. Additionally, the return air temperature at the outlet of the heat exchanger increased by 2 °C, which shows the effect of mixer arrangement on the heat exchange.

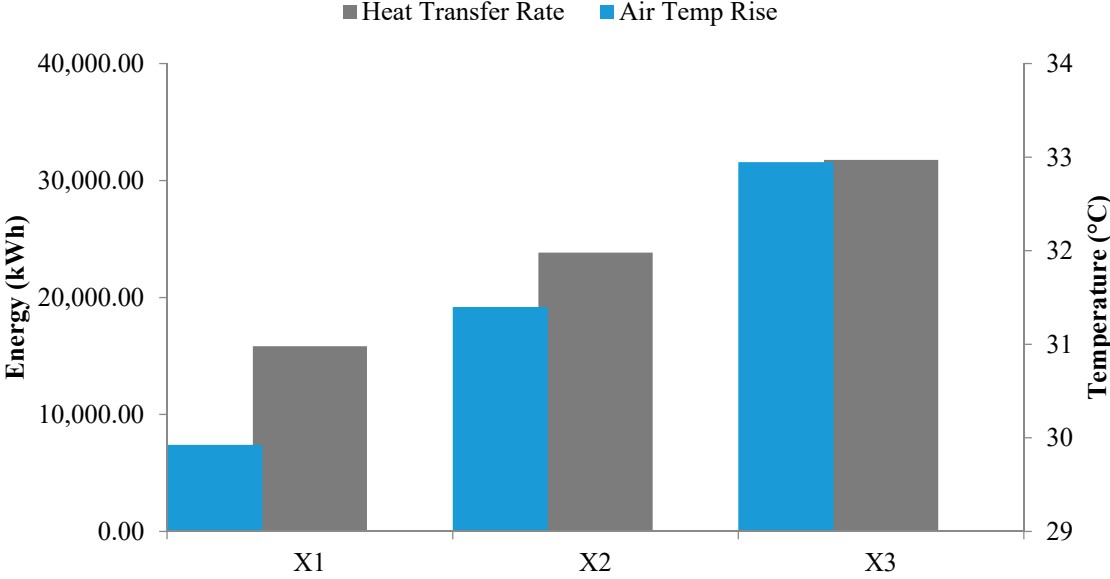

**Figure 11.** Energy rates and temperature of several diverter mixing ratios.

### 3.8. Optimal System Performance

After optimizing the parameters, the system was modeled to determine the moisture removal as well as the temperature rise of the fresh air entering the desiccant wheel represented by humidity ratio (HR) and temperature shown in Figure 12 for three sample days. It can be observed that the HR decreased from 0.024 to 0.016, while the temperature increased from 42 °C to 50 °C at the outlet of the desiccant wheel.

### 3.9. System Energy Calculation

To calculate the final energy savings from the system, the following energy equation was used:

Q = m × H
M = mass of moisture removed
H = enthalpy of water vapor = 2260 kJ/kg
$Q = \sum((w_{out} - w_{in}) \times 0.125 \text{ (h)}) \times H$
$Q = \sum((1130.64 - 890.11) \times 0.125) \times 2260$
Q = (141.33 − 111.26) (kg/kg).(h) × 2260 kJ/kg
Q = 67949.32 kWh

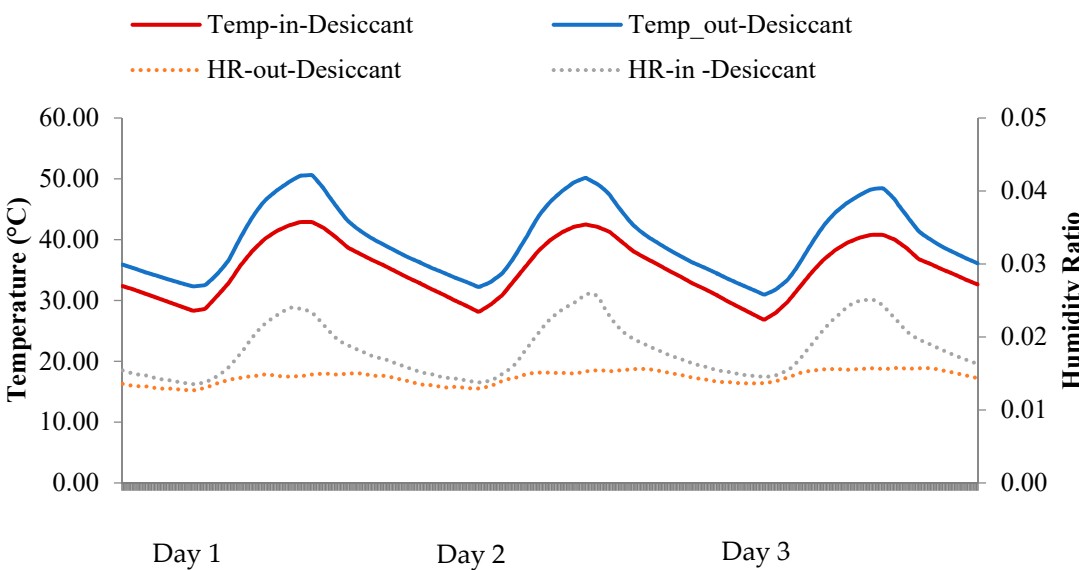

**Figure 12.** Humidity ratio and temperature in and out of the desiccant system.

Thus, the total energy savings of the desiccant system was 67,949.32 kWh. When we compared the total energy savings of the desiccant system to the initial cooling consumption rate of the school, which was 777,607.7 kWh, it showed that the system was saving around 10% of the conventional cooling system. Table 4 summarize all possible parameters influencing the system overall performance.

**Table 4.** Summary of parameters influencing the system performance.

| Parameter Name | Symbol | Range of Values | Equation Relevance |
|---|---|---|---|
| Flowrate | - | 1000–5000 kg/h | - |
| solar concentration Ratio | CR | 1–3× | 1 |
| Fluid inlet/outlet Temperatures | Hx1T–Hx4T | 75 °C–90 °C | 4 |
| Collector Energy | Coll-Energy | 18,000 kWh | 5 |
| Humidity Ratio | HR | 0.024–0.016 kg/kg | - |
| Heat Exchanger Effectiveness | HX | 0.5–0.8 | 7 |
| Diverter Mixing Ratio | - | 0.1–0.4 | - |

## 4. System Integration

The school is being served by a chilled water-cooling system with variable air volume arrangement. A solar thermal array is in place to meet hot water demand for swimming pool showers during winter. The solar desiccant system will work in combination with the chilled water system on one hand to dehumidify the supply air and the solar thermal system on the other hand to regenerate the desiccant wheel. The most promising aspect of the system integration lies in the capacity utilization of the installed thermal collector. The thermal collector will heat the cold service water in winter, while instead of being redundant in summer (as there is no hot water demand) it will be utilized to heat the return air to regenerate the desiccant wheel. The inlet fresh air (process air) passes through various channels of rotary dehumidifier, thus being dried by the desiccant material. The desiccant wheel being warmer increases the air temperature at the outlet of the wheel that is subsequently cooled by the cooler return air through the heat recovery wheel. The supply air is further cooled by the air conditioning system and served to space. The return air, after passing through the heat recovery wheel, is heated by the solar thermal collector through heat exchangers and passed through the desiccant wheel to regenerate the wheel. The hot and humidified air exits the desiccant wheel to the ambient air, as shown in Figure 13.

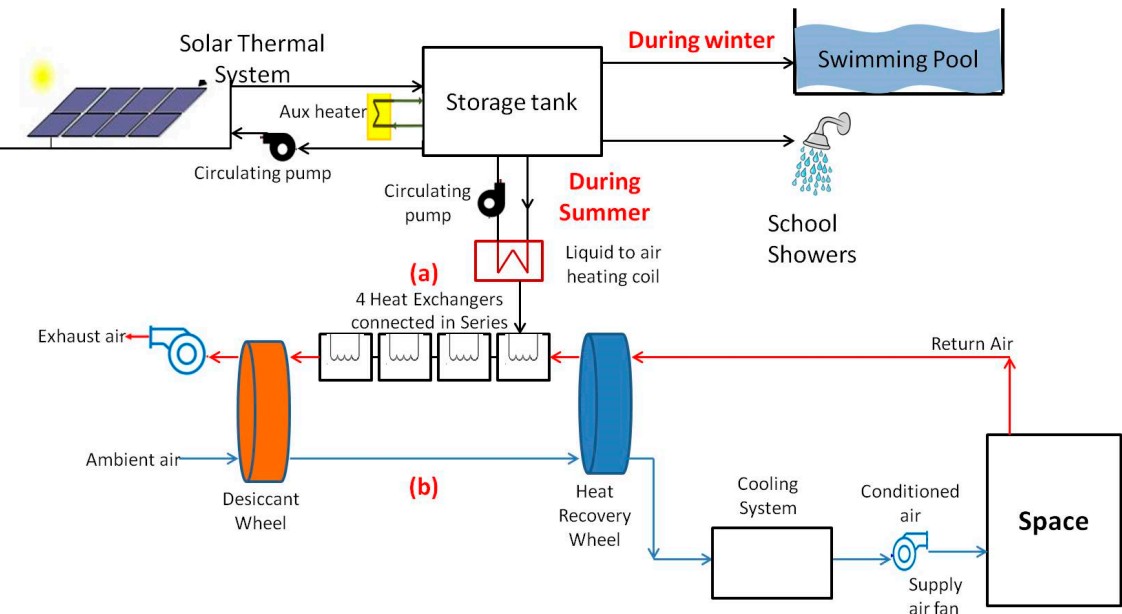

**Figure 13.** System Integration of solar desiccant dehumidification showing the return air stream (**a**) and supply air stream (**b**).

*Cost–Benefit Analysis of the System*

The cost–benefit analysis was conducted by calculating a simple payback period considering the benefit of energy saved for air dehumidification through the desiccant system compared to the existing condensing coil system and the cost of the systems (Table 5). The energy-saving was converted to monetary benefit by applying UAE-based electricity tariff of AED 0.32/kWh. The solar collectors being already installed in the school due to code requirements were not included in the system. The cost of solar concentration, inclusive of the tracking system and lenses was counted as $0.36 per watt installed, considering the cost of an equivalent CPV system. Energy consumed by the pump to circulate water through the collector was calculated by assuming 10 m additional water head with a flow rate of 0.18 L/s. It resulted in the additional shaft power of 30 W for pumping.

**Table 5.** Cost estimation of the set-ups.

| Component | Price/Unit |
|---|---|
| Pumps | $408 |
| Desiccant dehumidifier Desiccant Rotors International Pvt. Ltd. | $1530 |
| Heat exchanger Guangzhou Jiema Heat Exchange Equipment Co., Ltd. | $333 |
| Evaporative cooler Australian Construction Handbook, 2011 | $660 |
| Solar thermal collector Alizadeh, 2008 | $200/30 m$^2$ |
| Fans Australian Construction Handbook, 2011 | $286 |
| Air distribution units Australian Construction Handbook, 2011 | $1650 |
| The lifespan of system Baniyounes et al., 2012 | 25 years |

Source: http://apvi.org.au/solar-research-conference/wp-content/uploads/2015/12/Y-Ma_Peer-Reviewed_FINAL.pdf.

The total system cost = 10,867$.
The net benefit = 67,949.32 kWh ∗ 1.2 (unsubsidized international rate) = 81,539.184 kWh.
The payback period = 81,539.184 kWh /10,867$ = 7.5 years.

## 5. Conclusions

This paper presented an analysis of desiccant cooling systems in terms of the feasibility of replacing an existing condensing coil dehumidification system in a school building in

Abu Dhabi with a solar-assisted desiccant dehumidification system. The system was evaluated for its technoeconomic competitiveness, through simulation employing TRNSYS software based on energy savings compared with the existing system. For each tested parameter the temperature, energy rates, relative humidity, and humidity ratios were predicted. The temperature required for desiccant regeneration was achieved with the concentration ratio of 3×. The most suitable heat exchanger effectiveness was found at 0.6, which maintained the desired fluid temperature at the collector outlet. The optimal mixing ratio of four heat transfer fluid streams entering each of the four heat exchangers in an arrangement of (i) decreasing order of 0.4, 0.3, 0.2, and 0.1; (ii) constant ratios of 0.25, 0.25, 0.25, and 0.25; and (iii) increasing order of 0.1, 0.2, 0.3, and 0.4 were simulated. The solar thermal collector helped remove 35% moisture from fresh air being supplied indoors when it was coupled with a desiccant dehumidification system. The solar thermal collector provided 65% of the thermal energy required to dry out the desiccant wheel through return air while the rest of the 35% heating was provided by an auxiliary heating system integrated into the hot water storage tank. The proposed extension of the thermal collector to desiccant dehumidification can save 10% cooling energy in the building during summer as an added value without compromising its primary function, like a hot water production system in winter. Finally, the cost–benefit analysis comparing the pure energy benefit with the cost of the system installed revealed a payback period of around 7.5 years. In conclusion, solar-assisted cooling desiccant dehumidification is technically feasible for the hot climate of the UAE buildings and can significantly contribute to reducing Greenhouse Gas Emissions; however, it requires further improvement to render it financially attractive with a targeted payback period of below five years.

**Author Contributions:** Data curation, A.H.; formal analysis, K.A.T.A. and A.H.; investigation, J.A.D.; methodology, J.A.D.; writing—original draft preparation, K.A.T.A. and A.H.; writing—review and editing, K.A.T.A. and A.H. All authors have read and agreed to the published version of the manuscript.

**Funding:** Emirates Centre for Energy and Environment Research, United Arab Emirates University: 31R102.

**Institutional Review Board Statement:** Not applicable.

**Informed Consent Statement:** Not applicable.

**Data Availability Statement:** Not applicable.

**Conflicts of Interest:** The authors declare no conflict of interest.

## Abbreviations

| | |
|---|---|
| Aa | Aperture area (m$^2$) |
| B | First order angle modifier (θ) |
| b | Beam radiation |
| C | Film coefficient |
| CR | Concentration ratio |
| D | Second order angle modifier (θ) |
| d | Diffused radiation |
| DX | Dehumidification |
| DW | Desiccant wheel |
| $F_R$ | Overall collector heat removal efficiency factor |
| H | Enthalpy of water vapor (kJ/mol) |
| HR | Humidity Ratio |
| HX | Heat exchanger |

| k | Thermal conductivity (W/(m·K)) |
|---|---|
| M | Mass of moisture removed (kg) |
| s | Absorber width (m) |
| n | Number of internal reflections |
| T | Temperature (K) |
| PV /t | Photovoltaic/ thermal |
| TMY | Typical meteorological year |
| U | Heat transfer coefficient (W/(m$^2$K)) |

**Symbols**

| | |
|---|---|
| $\alpha$ | Collector absorptance |
| $\varepsilon$ | Heat exchanger effectiveness |
| $\theta_c$ | Half acceptance angle (degree) |
| $\tau$ | Transmittance |
| $\tau\,\alpha$ | Module transmittance absorptance product |
| $\rho_{eff}$ | Effective reflectance |
| $\bar{x}$ | CPC radius at truncated diameter (m) |
| $_R$ | Wall reflectance |
| $A_s$ | Focus area (m$^2$) |
| $C_{max}$ | Maximum heat capacity ratio of heat exchanger |
| $C_{min}$ | Minimum heat capacity of heat-exchanger |
| $d$ | Characteristic diameter of storage tank (m) |
| $h\circ$ | Convective heat transfer coefficient (W/(m$^2$K)) |
| $I_b$ | Radiation at modified angle (W/m$^2$) |
| $I_{in}$ | Incident Radiation (W/m$^2$ |
| $I_R$ | Reflected radiation (W/m$^2$ |
| $I_L$ | Radiation loss (W/m$^2$) |
| $I_T$ | Transmitted radiation (W/m$^2$) |
| $\dot{m}$ | Flow rate at use conditions (m$^3$/s) |
| $N_S$ | Number of identical collectors in series |
| $Nu_D$ | Local Nusselt number |
| $Q_u$ | Thermal energy gain (kWh) |
| $T_a$ | Ambient temperature (K) |
| $T_{i,j}$ | Outlet temperature (K) |
| $U_L$ | Tank bottom loss coefficient (W/(m$^2$K) |
| $W_{in}$ | Inlet water flow rate (m$^3$/s) |
| $W_{out}$ | Outlet water flow rate (m$^3$/s) |

## Appendix A. The TRNSYS Model of the Proposed Desiccant Dehumidification System

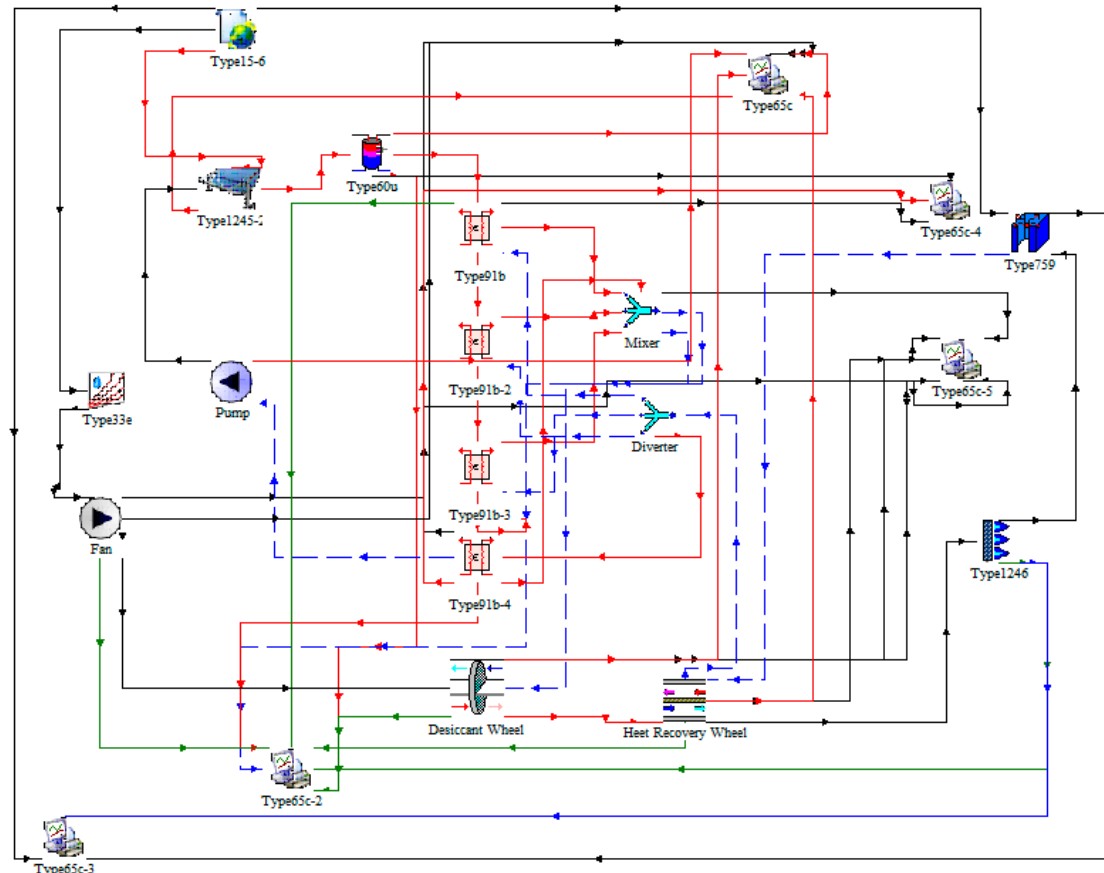

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
