# Peer review of "Assessment of Solar Dehumidification Systems in a Hot Climate"

_sustainability, doi:10.3390/su13010277_

Round 1

Reviewer 1 Report

The introduction part does not provide sufficient information. Literature review is too short in my opinion. The power of renewables installed on the presented building was not described. How big is the field of solar collectors? How much space there is on the roof for mounting any concentrating system?

The paper presents a dynamic model of dehumidification system based on concentrated solar collectors. The authors simulate the described system in TRNSYS software - Section Methodology. But there is no methodology, only a picture taken from the software. It is crucial to describe HOW it was simulated. Any equations, schemes of operation, controlling strategy - it has to be described.

In section 3 it is written that in winter months, triple concentration is sufficient for achieving 80-celsius degrees. I am sure, that no concentration is also a sufficient ratio - it can be achieved by lowering the flowrate of the working medium. But in the whole paper, there is no word about flowrate or other collector parameters. Another question - it is written, that in winter months with 3CR concentration it is possible to achieve 80-celsius degrees. But what will happen during the summer? Will the collectors operate on steam? 

Lines 230-236, I do not think that this style meets this journal criteria.

Table 3 - Citation style does not meet this journal criteria.  

I can not say anything about conclusions, because the paper does not provide enough information about the system and simulation. 

Author Response

Dear Reviewer

The authors would like to extend sincere thanks for the time and effort put in by the reviewer to improve the article. The authors have addressed comments and added corrections accordingly which greatly helped improve the paper. 

Best Regards

Authors

Reviewer 2 Report

General impression.

The article is difficult to follow as it does not provide the equations that are being implemented. This must be done, even if the generated system of equations is solved with TRNSYS. In addition to this, the charts are not sufficiently explained according to a study methodology and parameters, which must be presented in equations. As result, it is difficult to capture interest.

These deficiencies imply that the content / knowledge provided by the article in its current form is very limited.

Specific points:

Lines 55 and 57. There are missed the references in the References section: [9] and [10].

  • Could you enter a reference on line 60 for the statement: "nanofluids for better applied heat transfer in a greenhouse building in Saudi Arabia", this is important to know since it is commented in the text?
  • Are the models used from reference 11? If yes, present the fact and explain the implementation in TRNSYS using the required equations (including the reference where necessary).
  • In line 63-64, could give a reference. “A solid desiccant dehumidification employing photovoltaic thermal (PV/T) for thermal energy supply with Maisotsenko coolers…”is it from ref 11? This it is confusing.
  • Could you explain in the context of the present paper the statement:” The desiccant evaporative cooling system lowered the average daily maximum temperatures in the greenhouse by 6 °C compared to conventional evaporative cooling system?”
  • Line 66. The Reference [12] is also been lost in the References section.
  • Line 95: Are Data simulated or collected by measures? (Confusing) “Typical Meteorological Year (TMY) was collected from measurements from the school?” If collected or simulated how was it done?
  • Line 114 Describe deeper the equipment used and its characteristics by in tables and text.
  • Figure 131, please introduce the equations used for the desiccant wheel and heat recovery wheel (this is linked with a previous comment).
  • Line 144,”without solar (CR=1x…3x) the water temperature remained below 60 °C” and Fig 5. Explain this figure, parameters on it and implications in more detail.
  • Figure 6 and 7 difficult to understand as it is, in part because there is no expressed model of equations.
  • In line 182 are the values obtained from experimental measurements?
  • Finally the discussion respect the findings of other authors is missed.

Author Response

(The authors gave the same response as above.)

Reviewer 3 Report

The article attempts to investigate the feasibility if solar thermal Systems based on energy intensive condensing coil dehumidification system in building retrofitting. The topic is really interesting, and under investigated. The contents of the paper are not summarized in the abstract that is not comprehensive of your research. Can you rewrite it according to your methodology and results? The introduction is focused both on the building sector in Abu Dhabi and in the specific technology. You refer to few papers in your topic. You can stress the idea that only few papers are realized, demonstrating they this topic is under analyzed. Solar assisted desiccant dehumidification systems were realized also in other Countries, for example in Italy, as demonstrated also by the following paper I suggest to refer: 10.1016/j.egypro.2017.09.387 to enlarge your references. The aims of the paper are not clear. Add it, demonstrating the gap the literature review. A concise section on aims, steps and methodology could help in the comprehension of the paper. Particularly,  The part on the methodology may be implemented inserting the different phases of the work. The description of the case study is clear. The description of the simulation need for her explications, to understand better the conditions, the input parameters and the validation process. Concerning to validation, it is useful to refer to this reference:https://doi.org/10.1016/j.buildenv.2020.107081. Here you can find some input data to verify the results of you simulation according to the real behaviorist of the building. The discussion of your work is synthetic but  clear. Tables and description are clear. Conclusions must be improved with the most important finding of your research. Now are too general. Bullets points can help you.

Author Response

(The authors gave the same response as above.)

Round 2

Reviewer 1 Report

I suppose that the corrections made by the authors are sufficient to publish a paper in this journal. Thank you for responding to this review.

Author Response

The Authors are thankful for the time and effort from the reviewer to help improve the paper.

Reviewer 2 Report

The study is difficult to follow and more work is needed to improve the communication of ideas.

Some specific points are:

 1) It is recommended to draw a simple diagram, because the representation of TRNSYS at the introduction is difficult to follow according the explanation; 2) the new formulas introduced have several variables not defined either in the text or in a general nomenclature. Some variables are cited in the text, but do not appear in the equations; 3) the correlations of the coefficients of the film must be accompanied by their reference; 4) the parametric study being addressed and its justification should be presented before discussing the results. It is also suggested to use tables showing the values of properties ​​that remain constant and those that are studied in a parametric way.

In general, I consider that the paper has to be more elaborated (worked) in contents and structure, as well as when introducing the formulas and results.

Author Response

The Authors are thankful for the time and effort from the reviewer to help improve the paper. All the comments have been treated and a point by point response is added. 

Reviewer 3 Report

Thank you very much for the corrections. You improve a lot your paper. Now, the introduction is focused both on the building sector in Abu Dhabi and to the specific technology, stressing the need for more research. The reference 21 (10.1016/j.egypro.2017.09.387) refers to Italy in the text, but not in the reference, where it is cut. Please correct it. Some information on the input parameters and the validation process were added.

Author Response

The Authors are thankful for the time and effort from the reviewer to help improve the paper. The suggested change has been made to the paper. 

Round 3

Reviewer 2 Report

Despite successive revisions with several improvements, in the article there are important defects in the formulation of the model that make the general impression of the article poor and that it does not present the required level.

The definition of the nomenclature is not rigorous or complete, and units are also missing. There are several undefined variables in the text or in the nomenclature as for example in Eq. 1: x Ì…, θc; in Eq. 4: Ns, AFR, Ta; in Eq. (5) variables b and d; etc.; in Eq. (6), ho has to be multiplied simply by k and not to put it in parentheses, which is not understood.
It is suggested that a book like the one cited in the reference be taken as a reference [23].

Author Response

Dear Reviewer,

The authors are extremely thankful for your rigorous review and appreciate the time and effort spent to help improve the paper. We have addressed all the comments as advised and would be thankful in advance for your review and feedback. A point by point response sheet to your comments has been uploaded. 

Best Regards

Corresponding Authors
